# Cancer and Cardiovascular Disease: The Conjoined Twins

**DOI:** 10.3390/cancers16081450

**Published:** 2024-04-09

**Authors:** Mohammad Zmaili, Jafar Alzubi, Motasem Alkhayyat, Almaza Albakri, Feras Alkhalaileh, Joshua Longinow, Rohit Moudgil

**Affiliations:** 1Robert and Suzanne Tomsich Department of Cardiovascular Medicine, Sydell and Arnold Miller Family Heart, Vascular and Thoracic Institute, Cleveland Clinic Foundation, Cleveland, OH 44195, USA; zmailim@ccf.org; 2Department of Medicine, Division of Cardiology, Einstein Medical Center, Philadelphia, PA 19141, USA; 3Department of Gastroenterology, Hepatology and Nutrition, Digestive Disease and Surgery Institute, Cleveland Clinic Foundation, Cleveland, OH 44195, USA; 4Jordanian Royal Medical Services, Department of Internal Medicine, King Abdullah II Ben Al-Hussein Street, Amman 11855, Jordan; 5Department of Internal Medicine, Cleveland Clinic, Cleveland, OH 44195, USA

**Keywords:** cardiovascular disease, cancer, cardio-oncology, repurposed pharmacotherapeutics

## Abstract

**Simple Summary:**

Cardiovascular disease and cancer are two major causes of morbidity and mortality worldwide. The significant overlap between cardiovascular medicine and oncology led to the emergence of the cardio-oncology field. Understanding the pathophysiological basis of the interconnected relationship between cardiovascular disease and cancer is fundamental to improving patient care and clinical outcomes. Several cardiovascular therapies have proven beneficial in the oncologic field, and thus, may need to be incorporated into the therapeutic armamentarium against cancer.

**Abstract:**

Cancer and cardiovascular disease are the two most common causes of death worldwide. As the fields of cardiovascular medicine and oncology continue to expand, the area of overlap is becoming more prominent demanding dedicated attention and individualized patient care. We have come to realize that both fields are inextricably intertwined in several aspects, so much so that the mere presence of one, with its resultant downstream implications, has an impact on the other. Nonetheless, cardiovascular disease and cancer are generally approached independently. The focus that is granted to the predominant pathological entity (either cardiovascular disease or cancer), does not allow for optimal medical care for the other. As a result, ample opportunities for improvement in overall health care are being overlooked. Herein, we hope to shed light on the interconnected relationship between cardiovascular disease and cancer and uncover some of the unintentionally neglected intricacies of common cardiovascular therapeutics from an oncologic standpoint.

## 1. Introduction

### 1.1. Burden and Epidemiology of Cardiovascular Disease and Cancer

In spite of the remarkable improvement in cardiovascular outcomes over the last several years, cardiovascular disease (CVD) continues to be the number one cause of morbidity and mortality worldwide [1,2]. Additionally, cancer is also a leading cause of death globally, with a significant economic impact [3]. The most recent report from the Centers for Disease Control and Prevention (CDC) showed that in 2019, CVD and cancer were the leading causes of death in the United States, with 659,041 and 599,601 deaths, respectively [4]. The World Health Organization (WHO) estimates that 17.9 million people die every year from CVD and nearly 9–10 million die from cancer [5,6]. While the disease processes are an independent cause of increased mortality, recent evidence suggests that they are intertwined and therefore, mitigation of risk factors from one (such as CVD) can protect patients from the other (such as cancer) [7]. In this review, we highlight the significant points of intersection between CVD and cancer and discuss the major clinical implications in regard to prevention and management. 

### 1.2. Significance of Co-Occurrence of Cardiovascular Disease and Cancer

CVDs including coronary artery disease, carotid artery disease, peripheral vascular disease, cerebrovascular disease, and heart failure co-exist in more than 40% of patients with lung cancer, 30% with hematologic malignancies, 35% with renal cancer, 25% with head and neck cancers, 25% with colon cancer, 15% with breast cancer [8]. As cancer survivorship continues to improve with the advent of new therapies, an increasing number of cancer survivors are being followed by general practitioners [9]. As will be discussed in this review, these patients have a higher risk of cardiovascular complications and diseases, apart from the cumulative risk and increased prevalence of cardiovascular disease with aging, which in turn affects their long-term prognosis, and may even encumber their candidacy for cancer therapies. The realization of this association calls for a multidisciplinary approach with the incorporation of cardiovascular healthcare into the management and follow-up of cancer patients [10]. The aim of the cardio-oncology field is to ensure comprehensive medical care for the cancer patient, with an emphasis on risk stratification, complication prevention, and cardiovascular health optimization, which may collectively dictate eligibility for and response to various cancer therapies, and thus, overall outcome and survival [11]. 

The occurrence of CVD in cancer patients, not only affects long-term survival, but may also increase the susceptibility to cardiotoxic effects of certain therapies necessitating suboptimal dosage administration, or even worse, premature treatment cessation [12,13]. This has important ramifications as an analysis published by Copeland-Halperin et al. in 2020 showed that holding trastuzumab for 6 weeks or longer -owing to cardiotoxicity- increases the risk of invasive breast cancer recurrence or death, with an adjusted hazard ratio (HR) of 1.56% (95% CI, 1.10–2.21), in patients with early-stage human epidermal growth factor receptor 2 (ERBB2)-positive breast cancer [14]. It is clear that CVDs and their risk factors are pivotal predictors of cardiotoxicity associated with cancer therapy. More recent studies have shown that CAD, hypertension, and diabetes are considered strong predictors of left ventricular dysfunction among patients receiving anthracycline chemotherapy, whereas CAD, obesity, and hypertension, increase the risk of left ventricular dysfunction in breast cancer patients receiving trastuzumab [15,16]. Similarly, pre-existing hypertension was shown to be the strongest predictor of resistant hypertension requiring interruption of cancer therapy in patients receiving anti-angiogenic targeted agents [17]. This transcends into newer therapies also as more evidence emerge that therapies such as immune checkpoint inhibitors can cause myocarditis, valvulitis, and increases incidence of atherosclerotic disease [18,19,20].

### 1.3. Shared Risk Factors and Pathophysiological Processes

Even though CVD and cancer may appear to be two distinct entities, there are multiple areas of intersection. The two conditions tend to overlap, not only in regard to risk factors, but also in some underlying pathophysiological processes (Figure 1). In a recent large retrospective cohort study evaluating the effect of presence of CVD on cancer incidence, it was found that CVD, particularly atherosclerotic disease, was associated with an increased occurrence of specific cancer subtypes compared with those without CVD [21]. Interestingly, the risk of malignancy in patients with CVD was noted to be cancer-specific, with higher risk of lung, bladder, liver, colon, and other hematologic cancers. However, those patients were also found to have a lower risk of other cancer subtypes like breast, ovarian, and uterine cancers. There is a growing level of evidence that suggests shared etiologic mechanisms, of which inflammation stands out as a pivotal contributor to the manifestation of both conditions [22]. Nonetheless, other notable factors may contribute to the relationship between both disease processes. 

The role of inflammation in CVD pathogenesis is well-established, and many of the known CVD risk factors such as hypertension, smoking, dyslipidemia, obesity, and insulin resistance, can trigger atherosclerosis and lead to CVD events by inducing inflammation via various pathways [23]. This includes a heightened expression of pro-inflammatory cytokines, which leads to intensified oxidative stress due to increased production of reactive oxygen species and lipid peroxidation [23,24].

On the other hand, our understanding of the interaction between the immune system, inflammation, and cancer development is expanding, as many tumors have been shown to arise from sites of infection, chronic irritation, and inflammation [23,25,26]. Whether promoted by viral infections, smoking, or carcinogenic chemicals, inflammation can lead to cancer formation via complex pathways that include, whether directly or indirectly, enhanced cellular proliferation, dysregulated inflammatory response, amplified release of various cytokines, recruitment of inflammatory cells, or increased production of reactive oxygen species, which eventually can culminate in oxidative DNA damage, and disruption of DNA repair [23,27].

Other mechanisms that may explain the shared risk factors of cancer and CVD include hormonal-mediated disturbances seen obesity, diabetes mellitus, and physical inactivity [28]. In obesity, there is an increased expression of proinflammatory cytokines and hormones that are produced within adipose tissue, such as interleukin-6 (IL-6), tumor necrosis factor-alpha (TNFa), leptin, angiotensinogen, resistin, and C-reactive protein [29,30]. These molecules are believed to promote a steady state of low-grade inflammation and oxidative stress through formation of reaction oxygen species, which in turn may lead to DNA disruption, and eventually a higher risk of cancer development [31,32,33]. Additionally, leptin, which is secreted by adipose tissue, plays a major role in obesity-r,elated CVD [34]. IL-6 and TNFa can also induce hypertension and atherosclerosis, both of which play pivotal role in heart failure with preserved ejection fraction pathogenesis [35,36]. In this context, the CANTOS trial, which was randomized double-blinded study, investigated the effects of canakinumab, monoclonal antibody to proinflammatory cytokine IL-1β, in patients with prior myocardial infarction and elevated serum level of high high-sensitivity C-reactive protein (≥2 mg/dL) [37]. It was found that patients treated with canakinumab had a significantly lower rate of recurrent cardiovascular events compared to placebo group. But more interestingly, canakinumab was associated with a lower incidence of lung cancer, which further support the central role of inflammation in cancer and CVD pathogenesis [38].

Numerous studies have linked diabetes to cancer risk and its progression [39]. In diabetes, insulin resistance triggers atherosclerosis through oxidative stress, glycosylation, and high triglyceride levels, leading to endothelial damage in the vascular beds and thus atheroma formation [40,41]. Similarly, diabetes promotes a pro-inflammatory state that can mediate and increase the risk for cancer development and progression [29,42,43]. Additionally, sex hormone disturbances, which may result from hyperinsulinemia in the setting of disturbed glycemic homeostasis, are implicated in the carcinogenesis of some tumors such as breast and endometrial cancers [42,44,45,46]. Moreover, insulin-like growth factors, which are multifunctional peptides that regulate cell proliferation, differentiation, and apoptosis, are increased in the setting of impaired glycemic regulation and insulin resistance. This in turn can, not only enhance tumorigenesis and promote the development of some cancers, but also mediate smooth muscle proliferation in blood vessels with resultant atherosclerosis [39,47,48,49,50]. Correlation exists at the molecular level also as recent studies showed that potency of immune checkpoint inhibitors are greatly enhanced by PCSK9 inhibitors [51].

Furthermore, several studies have showed that metabolic reprogramming play a significant role in both heart failure and cancer molecular pathogenesis [52]. This metabolic shift at a cellular level occurs in tumors cells and failing cardiomyocytes in response to various stressors. Specifically, healthy myocardium mainly relies on beta oxidation of fatty acids for ATP synthesis [53]; however, metabolic source of energy in cardiomyocytes switches away from fatty acid utilization to glucose, ketone bodies, and amino acids (such as glutamine) under stress like pressure overload conditions [54,55]. This metabolic switch in failing cardiomyocytes leads to production of metabolites that are essential for biosynthesis as well as ventricular hypertrophy and contributes to calcium mishandling and ultimately cardiac dysfunction [42,56]. A similar shift in metabolic dependency is seen in cancerous cells to meet the catabolic and anabolic needs of growing tumors where the source of energy of proliferating cells shifts to glucose consumption “Warburg effect” along with preferential production of lactate, even in the presence of adequate tissue supply of oxygen, unlike healthy tissues that depends mainly on fatty acid oxidation for its energy expenditure [57,58]. This metabolic alteration in cancer cells is thought to support de novo synthesis of nucleotides, lipids, and proteins that are needed for tumor growth. [57] It is important to understand that the effects of these metabolic alteration in cancers and failing hearts extend beyond just altering source of energy but also have pleiotropic effects mediated by the byproducts of the metabolic changes and are also directly linked to organ dysfunction [59] In addition, certain oncometabolites produced by cancers, such D-2-hydroxyglutarate and succinate, have been directly implicated in development of cardiomyopathy in animal studies [60,61].

## 2. Cardiovascular Health in Cancer Patients

### 2.1. Role of Cardiovascular Disease Screening

The importance of primary prevention to reduce the overall burden of cardiovascular disease cannot be overemphasized. The first message of the 2019 American College of Cardiology/American Heart Association (ACC/AHA) Task Force on Clinical Practice Guidelines Report states that promotion of healthy lifestyle is the most important measure to reduce and prevent cardiovascular disease [62]. Additionally, the report lays out detailed recommendations to determine an individual’s risk for cardiovascular disease and to facilitate decisions in regards to preventive strategies and therapies. Several measures have been adopted to estimate such a risk, utilizing clinical tools and data such as the 10-year atherosclerotic cardiovascular disease (ASCVD) risk, the SCORE (Systematic COronary Risk Evaluation) project, among others [63,64,65,66,67,68]. The ultimate goal is to identify patients who are at a high risk of cardiovascular disease to a point where the benefit of a preventive intervention such as statin prescription surpasses the risks of possible adverse effects. As discussed, cancer patients represent a vulnerable population with an elevated risk of cardiovascular disease given the inherent properties of the risk factors that are shared among both conditions [28,69]. In various types of cancer, cardiovascular disease and mortality were shown to be greater in cancer survivors compared to age-matched controls [70,71,72,73]. Therefore, additional emphasis should be placed on risk assessment and reduction in this patient population. This may not only improve the overall cardiovascular health and outcome, but may also prove beneficial from an oncologic perspective. The development of cardiotoxicity has been reported to adversely affect outcomes of patients with cancer, and its prevention may be beneficial to overall patient survival [74,75,76]. Thus, it is fundamental to involve a cardiologist in the care of cancer patients, not only to detect early cardiovascular side effects of cancer therapy, but also to optimize the overall cardiovascular care of cancer patients from the initial cancer diagnosis to survivorship.

### 2.2. Cardiovascular Disease Screening in Cancer Patients. Where We Stand, and What Are the Implications?

While it has become more evident that cardiovascular disease and cancer share several risk factors, and therefore tend to commonly coexist, preventive measures have not converged as the two conditions are generally approached independently. The American Society of Clinical Oncology (ASCO) advocates for early implementation of routine cardiovascular surveillance in high-risk cancer patients, along with cardiovascular risk factor screening and modification [77]. Similarly, the American Heart Association (AHA) delineated a significant interrelation between cardiovascular disease and breast cancer, and provided a comprehensive review of the substantial areas of overlap [78]. The diagnosis of cancer often leads to mental stress, anxiety, and significantly impacts patients’ lifestyle. The influence of cancer detection affects medical providers as well. The focus granted to cancer management and prognosis often shifts the attention away from cardiovascular health promotion, and therefore, the opportunity for risk factor screening and modification is inadvertently missed [79]. The untoward sequelae of this cognitive distraction are expected to become more substantial as advancements in cancer detection and therapies have improved cancer survival rates [80]. This in turn is expected to increase the burden of cardiovascular disease, either as a complication from cancer therapy or from the cumulative effect of risk factors with increased longevity [78]. Previous studies have shown that cancer survivors may have poor overall control of their traditional cardiovascular risk factors [81]. There are also significant numbers of cancer patients with established CVD diagnosis who do not get referred to cardiologists, and are thus suboptimally managed [8]. Even if prompt action is undertaken to estimate the cardiovascular disease risk, the commonly used clinical assessment tools tend to underestimate the risk in patients with cancer, whether active or in remission [82]. As an example, a study of 561 breast cancer patients assessed the coronary artery calcium score derived from computed tomography (CT) scans that were performed for radiotherapy planning, and found that one third of patients who demonstrated a high coronary artery calcium score lacked other cardiovascular risk factors [83]. As such, these patients would not have been classified in the appropriate risk category group by clinical risk assessment tools. The implication of a heightened cardiovascular disease risk translates into higher rates of cardiovascular mortality in cancer patients compared to the general population [84]. An observational population-based study scrutinized the rates of cardiovascular mortality among 28 cancer types in cancer survivors from 1973 to 2012. Results showed that 11.3% of deaths were attributable to cardiovascular disease. Interestingly, the risk of cardiovascular mortality exceeded that of cancer mortality in eight types of cancer in at least one calendar year [84].

## 3. Cardiovascular Interventions in the Armamentarium of Cancer Therapy

Given the similarities and significant overlap between cardiovascular disease and cancer as discussed above, it is not surprising that cardiovascular interventions may provide a benefit and be useful in the oncologic field. Significant observations were noted in the past, in regards to a beneficial effect on the natural history of cancer, from cardiovascular medications. This favorable role encompasses a range of effects, from providing a protective effect against cancer development and progression, to augmenting the antitumor actions of cancer therapeutics. Below is a brief overview of commonly prescribed cardiovascular medications from an oncologic perspective, which is also summarized in Table 1.

### 3.1. Aspirin

Aspirin is an irreversible inhibitor of cyclooxygenase-1 (COX-1) enzyme that produces precursors for prostaglandins and thromboxanes [85]. This results in dose- and time-dependent inhibition of thromboxane A2 (TXA2) formation, an important mediator in platelet recruitment and aggregation [86].

Aspirin is generally considered the mainstay antiplatelet therapy for the treatment of acute coronary syndromes as well as the secondary prevention of atherothrombotic events in patients with various atherosclerotic diseases [87]. A metanalysis of sixteen secondary prevention trials showed that treatment with low-dose aspirin is effective in preventing approximately one-fifth of atherothrombotic vascular complications in patients with previous myocardial infarction, stroke, or transient cerebral ischemia [88]. Nonetheless, aspirin use for primary prevention of cardiovascular diseases is still controversial with uncertain balance between cardiovascular benefits and bleeding risk [89].

In the last few decades, compelling data have emerged suggesting an association between the regular use of COX inhibitors, including aspirin and other nonsteroidal anti-inflammatory drugs, and reduced risk of colorectal cancers [90]. The protective effects of aspirin against cancer are believed to be related to the prevention of early neoplastic transformation in addition to an anti-metastatic action [91]. It is evident that platelets play a pivotal role in neoplastic transformation via enhanced biosynthesis of prostaglandin E2 (PGE2), among several lipid mediators that are synthesized and released by activated platelets. PGE2 influences the adhesive, migratory, and invasive behavior of cells and create an environment that facilitates tumor formation and progression [90].

Hence, multiple randomized controlled trials (RCTs) were designed to study the chemo-preventive effects of COX inhibitors, given the data that have suggested an important role of COX enzymes, particularly COX-2, in gastrointestinal carcinogenesis [92,93]. Nonetheless, some of these trials that were designed to evaluate these chemo-preventive effects of selective COX-2 inhibitors, such as celecoxib and rofecoxib, showed an increased risk of major adverse cardiovascular events and were halted [94,95]. On contrast to selective COX-2 inhibitors, RCTs, observational case-control and meta-analysis studies have demonstrated a chemo-preventive effect of aspirin against colorectal cancers [96,97,98,99,100,101]. Other studies have also demonstrated that aspirin reduced the risk of death from several non-colonic solid cancers including esophageal, pancreatic, brain, lung, stomach, and prostate cancer in patients with Lynch syndrome [101,102]. However, it is unclear whether there is a chemoprotective benefits against malignancies other than colorectal cancer in patients without Lynch syndrome. Nonetheless, the evidence of use of aspirin and its mitigating effect on colorectal cancer morbidity and mortality has been identified [103,104,105].

### 3.2. Angiotensin Converting Enzyme Inhibitors (ACEi) and Angiotensin Receptor Blockers (ARB)

Angiotensin converting enzyme inhibitors (ACEi) and angiotensin receptor blockers (ARB) are widely used in the management of hypertension and heart failure [106,107]. In addition, they have renal protective effects in patients with diabetes [108]. It is evident that the renin–angiotensin–aldosterone system (RAAS), including angiotensin II and angiotensin II type 2 (AT2) receptor subtypes, regulates blood pressure homeostasis and electrolyte balance [109,110]. The functions of RAAS extend not only to the cardiovascular system but also involve multiple organ systems such as the kidney, brain, pituitary, adrenal, gonad and adipose [111,112]. Previous research and human studies have demonstrated RAAS signaling within various organs and tissues, indicating its essential role in several biological processes with involvement in different pathophysiological mechanisms, including inflammation [113,114,115]. Additionally, other studies have shown that angiotensin II functions as a paracrine and/or autocrine signal in some cancers and mediates recruitment of inflammatory cells. This leads to an enhanced secretion of cytokines that accelerate cell proliferation and tumor angiogenesis, such as up-regulating vascular endothelial growth factor (VEGF) expression [113,116,117]. Furthermore, the RAAS may modulate cancer growth and progression at different levels, including sustained angiogenesis, evasion of apoptosis, self-sufficiency in growth signals, insensitivity to anti-growth signals, tissue invasion and metastases, and limitless replicative potential [117]. In preclinical models, RAAS inhibitors have shown efficacy in reducing metastases, whereas AT1 R expression frequently correlates with the degree of cancer invasiveness. In two separate lung metastases models, the treatment of mice with candesartan (ARB) significantly reduced lung metastatic burden [118,119]. whereas captopril (ACEi) significantly reduced cancer size and was associated with a decreased lymph node metastases in a non-small-cell lung cancer xenograft model [120]. Wherein human retrospective studies, they have provided some evidence that long-term use of RAAS inhibitors might modulate cancer growth and progression [121,122,123]. One of the first studies that assessed the risk of cancer in hypertensive patients who received ACE inhibitors over a 15-year period, showed that the relative risk of incident and fatal cancer among 1559 patients receiving ACE inhibitors was significantly reduced in comparison with control subjects, most markedly for female-specific cancers [124]. In three other retrospective case–controlled studies, ACE inhibitor use was associated with reduction in the incidence of esophageal, pancreatic, and colon cancer [117,125,126]. Other studies showed that the use of RAAS antagonism has also been associated with a reduced risk of melanomas and a lower risk of developing prostate cancer [121,122]. These retrospective studies suggested that dysregulation of RAAS components plays a role in a broad range of human malignancies and may correlate with disease outcome. More importantly, these studies present a large body of evidence that RAAS inhibitors play an important prognostic indicator and novel molecular target for a wide range of cancers. Other retrospective studies have reported that losartan (ARB) use prior to chemotherapy potentiates the anticancer response by improving chemotherapy delivery to cancer cells [127,128,129]. A more recent large meta-analysis of nine studies (*n* = 1362) that evaluated the effects of beta blockers, ACEi, and ARBs on cardiotoxicity of trastuzumab and anthracycline reported the those therapies were associated with the preservation of LVEF [130]. Although the link between RAAS, cancer angiogenesis and invasion arguably represent a therapeutic opportunity for clinical intervention, few clinical trials have been initiated to investigate the efficacy of RAAS modulators in cancer. This may be attributed mainly to the complex nature of RAAS signaling, making the response to RAAS inhibitors, either individually or in combination with other drugs, difficult to predict [117]. On the other hand, some observational studies have shown that the use of some RAAS inhibitors was associated with an increased risk of lung cancer [131].

### 3.3. Beta Blockers (βB)

Beta (β)-adrenoceptors are broadly distributed in various tissues and regulate wide range of important physiological functions and disease states [132]. β-adrenergic receptor blockers have been widely used for the treatment of hypertension, ischemic heart disease, and congestive heart failure [133,134]. Several studies have shown that catecholamines can significantly enhance the ability of tumor cells to invade the surrounding extracellular matrix via beta adrenergic system, thereby enhancing tumor growth through activation of invasive ability and stimulating VEGF secretion [135,136,137]. It has been evident that the β-adrenergic system plays pivotal roles in cancer development and progression and is involved in almost every step of cancer development, including stimulation of continuous proliferation along with evasion of growth suppressors, resistance to apoptosis, enhancement of invasion and metastasis, and induction of angiogenesis [137]. A translational study showed that norepinephrine and the β-adrenergic agonist isoproterenol can enhance the production of a proangiogenic cytokine, vascular endothelial growth factor, by ovarian cancer cells [138]. Interestingly, these effects were completely blocked by propranolol (βB), suggesting that β-adrenergic receptors mediate production of proangiogenic factors and thereby facilitation tumor metastasis [138]. Hence, the potential benefits of βB to alleviate the deleterious progression of cancers influenced by β-adrenergic system have been further investigated. A number of studies have evaluated the effect of βB use on cancer, but they have had conflicting or inconsistent findings. First, a large retrospective case-control study of prostate cancer patients, which investigated the effect of different classes of antihypertensives on cancer (including βB, ACEi, calcium channel blockers, alpha-blockers), showed that only βB use was associated with a significant reduction in prostate cancer risk [139]. In another observational study that looked at outcomes of patients with cardiovascular disease on a 10-year-follow up, it was noted that βB use was associated with a significant reduction in cancer incidence [140]. Other studies did not show or support protective benefits for βB against cancers [141,142]. Several prospective clinical trials that are assessing propranolol use in patients with ovarian (NCT01504126, NCT01308944), colorectal cancer (NCT00888797), breast (NCT01847001, NCT00502684, NCT02596867), and melanoma (NCT01988831) are under investigation. The result of these trials would illuminate the potential chemoprotective effects of βB against some cancers.

### 3.4. Calcium Channel Blockers (CCB)

Calcium channel blockers (CCBs) are one of the first-line treatments for hypertension that function by blocking T-type calcium channels [143]. These channels regulate calcium homeostasis that controls various cellular processes, including those relevant to tumorigenesis, such as proliferation, apoptosis, gene transcription and angiogenesis [144]. Thus, it is evident that increased T-type calcium channel expression and function have an important role in the abnormal proliferation of cells in many types of cancers [145]. Interestingly, it has been suggested that the expression of T-type calcium channels in cancer cells may vary depending on the rate of cellular proliferation. [145,146]. Calcium has the ability to function both as a promoter of cellular proliferation, and as an inducer of cell death, depending on the amplitude of the increase in the intracellular calcium concentration, and the duration of this change. Consequently, both activators and inhibitors of calcium channels may have potential anti-cancerous effects [146]. The role of T-type calcium channels in cellular proliferation has been described in breast, brain, colorectal, gastric, ovarian, and prostate tumors as well as leukemia [146,147,148,149,150].

### 3.5. Statins (β-Hydroxy β-Methylglutaryl-CoA (HMG-CoA) Reductase Inhibitors)

Statins are considered a first-line treatment for hypercholesterolemia and a cornerstone therapy for atherosclerotic diseases [151]. Statins suppress cholesterol synthesis via inhibition of 3-hydroxy-3-methyl-glutarylcoenzyme A reductase, a pivotal enzyme in cholesterol synthesis [152]. Some cholesterol precursor molecules, which are also inhibited by statin therapy, are essential components of other critical cellular functions including cell membrane integrity [153]. Therefore, suppression of these molecules may interfere with cellular growth and development of cancers. In addition to their effects on cholesterol, multiple in-vivo and in-vitro studies have shown antiproliferative effects on various types of cancers [154,155,156,157,158,159]. Most data on the potential effects of statins on cancers are derived from observational studies. Multiple studies and metanalysis have described a decreased risk of prostate cancer in association with statins use [160,161,162,163]. Nevertheless, there were conflicting data on effect of statins on prostate cancer progression with some studies showing no effect of statins on progression-free survival after radiotherapy for prostate cancer and radical prostatectomy [164]. While other retrospective cohort studies showed a reduced biochemical recurrence in prostate cancer patients treated with radical prostatectomy [165,166]. Similarly, chemoprotective effects of statins on gastrointestinal cancers were also extensively studied. For example, a large meta-analysis has reported a protective association between statin use and gastric cancer risk among both Asian and Western population, in a dose-dependent manner [167]. Additionally, several studies that examined the association between statin use and colorectal cancer risk and survival have reported conflicting results. In one case-control study, statin use was associated with lower risk of colorectal cancer but no significant association with colorectal cancer specific mortality was found [168]. In another study, statin use was not associated with reduced risk of colon cancer, but was associated with reduced risk of rectal cancer [169]. Moreover, studies that have evaluated the association between statin use and risk of breast cancer reported conflicting results. it appears that statin with lipophilic structures had more chemoprotective effect against breast cancers compared to hydrophilic ones [170,171,172]. Some data have also suggested that statins have a favorable effect, particularly in patients with triple negative breast cancer [173,174,175,176,177]. Furthermore, the effect of statins on response to systemic anticancer therapy in patients with solid cancers was evaluated in observational studies. In one meta-analysis, statin did not appear to improve response to cytotoxic therapy in patients with solid tumors [178]. In a recent RCT involving patients with lymphoma who were scheduled to receive anthracycline-based chemotherapy, the odds of cardiotoxicity after anthracycline treatment were approximately three times greater for patients randomized to the placebo compared with those randomized to atorvastatin [179]. Although statin therapy shows some benefits in several types of cancer, these chemoprotective effects need to be further evaluated with RCTs.

### 3.6. Proprotein Convertase Subtilisin Kexin Type 9 Inhibitors (PCSK9-i)

Proprotein convertase subtilisin kexin type 9 (PCSK9) is a proteolytic enzyme that plays an important role in hepatic cholesterol homeostasis [180,181,182]. Among its various functions, it is known for its role in the reduction of low-density lipoprotein (LDL) receptor expression on hepatocytes’ cell surface [183,184,185]. Downregulation of these receptors, which is mediated by intracellular signaling and targeting for degradation, leads to higher LDL plasma levels with resultant hypercholesterolemia and heightened cardiovascular disease risk [186,187]. As the understanding of PCSK9 role in the pathogenesis of hypercholesterolemia evolved, efforts devoted to this unique pathway led to the development of monoclonal antibodies against PCSK9 [188]. Two FDA-approved injectable human monoclonal antibodies (Alirocumab and Evolucumab) are now available for use, either as monotherapy or as an add-on to other anti-lipid therapies, in patients with familial hypercholesterolemia or in patients who have intolerance to traditional first-line therapies, i.e., statin therapy [189,190,191,192,193]. Both have shown remarkable reductions in LDL levels compared to placebo, with a range of approximately 30–70% reduction, and up to 60% reduction in patients who are already on statin therapy [194,195,196,197,198,199,200,201,202,203].

Similar to the effects on LDL receptors, PCSK9 has been increasingly recognized as a regulator for other cell surface receptors, some of which are key modulators in immune signaling and response [204]. Moreover, cholesterol was found to have inhibitory effects on the antitumor responses of CD8+ T cells, and it is also involved in major histocompatibility protein class I (MHC-I) recycling on cell membranes [205,206]. These pathophysiological observations provoked the hypothesis of PCSK9 being a regulator of anti-tumor responses and thus, a potential target to enhance immunity against tumors [51]. Eliminating the effect of PCSK9, whether via direct inhibition or gene knock-out, was shown to attenuate tumor growth in mice cancer cells by decreasing the barriers or checkpoints to T call signaling, which then boosts the anti-tumor immune response to cancer cells [51]. PCSK9 deficiency also increased the expression of MHC-I on the surface of tumor cells, exposing them to the immune defenses and allowing greater intra-tumoral cytotoxic T cell infiltration [51]. These anti-tumoral effects were most notable in the presence of immune checkpoint therapy with anti-programmed cell death ligand 1 (anti-PD1) agents, where a synergistic action has been demonstrated [51]. Furthermore, the introduction of small interfering RNA (siRNA) against PCSK9 into human lung adenocarcinoma cells downregulated anti-apoptotic molecules and induced mitochondrial dysfunction, which then hampered tumor activity by promoting cell apoptosis [207]. A recent study in Japan showed that higher levels of PCSK9 antibodies in the serum was associated with favorable postoperative prognosis in esophageal cancer patients compared to low antibody levels [208]. In another Italian pilot study of 44 elderly patients with advanced and pretreated non-small cell lung cancer, serum levels of PCSK9 below a certain cutoff (95 ng/mL) at the second nivolumab dose was associated with better overall survival in comparison to higher levels [209]. In an experimental mouse models of breast and colon cancer, anti-PCSK9 vaccine, which led to lower plasma level and activity of PCSK9, was associated with moderate but not significant tumor growth reduction and prolongation of lifespan [210,211]. In a prior observational Mendelian randomization study, genetic variants that simulate PCSK9 inhibition were associated with lower breast cancer risk [212]. This emerging data makes PCSK9 inhibitors a plausible consideration for future trials to evaluate their clinical effect as an adjunct therapies for various cancers, especially those in which immune therapy have proven efficacious.

### 3.7. Sodium-Glucose Co-Transporter-2 Inhibitors (SGLT-2 Inhibitors)

SGLT-2 inhibitors were initially developed for diabetes treatment. They work through inhibition of sodium-glucose co-transporter-2 that is responsible for active cellular uptake of glucose and sodium, thereby decreasing glucose reabsorption in the proximal convoluted tubules of the nephrons and causing glycosuria, which leads ultimately to lowering blood glucose level [213]. However, it became evident over the past decade that the benefits of SGLT-2 inhibitors extend beyond blood glucose control in diabetic patient as multiple studies have shown significant beneficial cardiovascular effects in patients treated with SGLT-2 inhibitors [214,215]. Several randomized control trials have shown cardiovascular benefits in heart failure patients. The most recent data from DAPA-HF (Dapagliflozin and Prevention of Adverse Outcomes in Heart Failure) [216] and EMPEROR-Reduced (Empagliflozin Outcome Trial in Patients With Chronic Heart Failure With Reduced Ejection Fraction) [217] showed reduction in heart failure hospitalizations or cardiovascular death when compared with placebo. More recently, EMPEROR-Preserved (Empagliflozin Outcome Trial in Patients With Chronic Heart Failure With Preserved Ejection Fraction) trial have reported that treatment with Empagliflozin in patients with heart failure and a preserved ejection fraction is associated with a reduced combined risk of cardiovascular death or hospitalization for heart failure in patients, regardless of the presence or absence of diabetes [218]. Furthermore, CANVAS study (Canagliflozin and Cardiovascular and Renal Events in Type 2 Diabetes) that combined data from two trials including more than ten thousands patients with type 2 diabetes and high cardiovascular risk, has showed that canagliflozin was associated with a reduction in major adverse cardiovascular events compared with placebo [219]. Similar results of cardiovascular benefits were replicated in other studies [220,221]. Interestingly, the cardiovascular benefits derived from SGLT2 inhibitors were independent of diabetes status of patients, which strongly suggest alternative mechanisms for the reported cardioprotective effects [221]. Several mechanism were proposed to explain the those cardioprotective effects such as decreased production of leptin and reduced pericardial adipose tissue deposition as well as inflammation [222,223]. In addition, SGLT2 inhibition was found to shift metabolism toward more lipid oxidation and ketone production along with reduced glucose oxidation [224,225]. other potential mechanisms for cardiovascular benefits of SGLT2 inhibitors include enhanced natriuresis, reduction in plasma volume and blood pressure as well as improvement of systemic endothelial function and arterial stiffness [226,227]. In cancer context, several meta-analysis studies have examined cancer risk in patients treated with various SGLT2 inhibitors, but they did not show significant chemoprotective effects [228,229,230]. Several translational studies have reported antiproliferative and chemoprotective effects with SGLT2 inhibition in certain types of cancers [231,232,233]. As described above, cancer cells exhibit metabolic reprogramming “Warburg effect” that promotes the survival and progression of cancers. Data from in-vitro studies showed amelioration of metabolic reprogramming seen in some cancers. For example, it was shown that SGLT2 inhibitor suppresses hepatocellular cancer (HCC) in-vitro growth through blockage of glucose influx-induced β-catenin action, which is a pro-oncogenic protein [234]. In another study, in-vitro treatment with SGLT2 inhibitors demonstrated disruption of adhesion capacity of certain cancer cells and suppression of oxidative phosphorylation via inhibition of mitochondrial electron transport chain in HCC, breast, prostate, and lung cancer cells [228,235,236].

### 3.8. Exercise

Before we conclude our discussion on the oncologic effects of cardiovascular interventions, it would be worth mentioning that exercise, which is thought of as a cardiovascular lifestyle intervention, can also boost cancer immunity. Current epidemiological evidence suggests that regular physical activity and exercise influence cardiovascular health in several ways. This include antiatherogenic effects in blood vessels, including improvements in vascular endothelial function and structural vascular adaptations, and a healthy autonomic balance (regular exercise increases vagal tone to the heart and prevents malignant arrhythmias) [237,238]. Additionally, regular exercise can also prevent fatal arrhythmias by inducing cardiac preconditioning, which provides a cardio-protective effect against ischemia-reperfusion injury [239,240,241,242]. Nonetheless, the cardiac preconditioning effects of exercise is still widely under-recognized and mostly evident in preclinical studies [243].

The benefits of exercise training for cancer patients are also becoming increasingly evident. It has been shown that aerobic exercise reduces cancer incidence and inhibits tumor growth [244,245]. Epidemiological studies have shown that physical activity reduces the risk of at least 13 different cancer types with an exercise-dependent reduction in the risk of disease recurrence for certain cancers [246]. Likewise, numerous preclinical exercise studies showed similar exercise-dependent protection against cancer [247]. Across the vast majority of preclinical studies investigating the effect of exercise on cancer outcomes, exercise has been shown to reduce the rate of tumor growth [244,245,247].

Additionally, exercise training not only can reduce tumor growth, but may also have the potential to augment the potency and efficacy of traditional cancer therapies [248]. As the efficacy of both chemotherapy and immunotherapy relies on adequate blood perfusion to the tumor, exercise training strongly affects blood circulation and oxygen delivery to peripheral tissues and thereby enhance delivery of the cytotoxic drugs to the interior of tumors [245,249,250].

## 4. Future Perspectives… Will We Be Ready to Change Our Protocols in the Near Future?

As elucidated, the interrelation between cancer and CVDs manifests through shared risk factors and molecular mechanisms. For many years, the field of cardio-oncology has predominantly focused on the development of CVD during or following cancer treatment. However, contemporary investigations have unveiled the potential for cancer to instigate or exacerbate CVDs, resulting in exacerbated prognostic outcomes.

Importantly, advancements in cancer therapeutics and increased longevity among cancer patients have brought to the fore a concomitant rise in the prevalence of CVDs within this demographic. Nonetheless, it is evident that even cancer patients with established diagnoses of cardiovascular diseases are less likely to receive standard treatments compared with those without cancer. Furthermore, many standard chemotherapy regimens are associated with direct cardiotoxic effects; many of which are unpredictable and associated with increased worse outcomes that account in part for increased mortality and morbidity in this cohort. Additional cancer- or cancer therapy-related factors that may complicate or undermine treatment of cardiovascular diseases including elevated risk of bleeding, thrombosis, and hypercoagulability in certain patients. For example, coronary artery disease in cancer patients may not be adequately treated, and they are less likely to undergo percutaneous coronary revascularization [251,252,253]. This reluctance may stem from concerns about or perceptions of an increased risk of stent thrombosis, or bleeding with optimal antiplatelet therapy. Therefore, it is crucial to study effects and safety of routinely used cardiovascular therapies in cancer patients, which may ultimately assist in development of cancer-specific risk scoring system that guide clinicians for treatment of cardiovascular diseases in cancer patients.

On management front, cardiovascular medications may be used as potential adjuvant therapies to standard anticancer treatments that could improve anti-tumor response and effectiveness. Although further studies are needed, repurposing cardiovascular therapies for non-cardiovascular indications and extending these interventions to cancer patients could be a promising therapeutic opportunity for patients with different stages of cancer who may not respond to standard cancer therapy or even lack an effective therapeutic treatment. Although preclinical studies have shown that these medications can demonstrate chemoprotective effects, or improve the effectiveness of standard chemotherapy regimens by modulating various pathways and molecular targets that are integral parts of tumorigenesis, the exact mechanisms by which some of the cardiovascular therapies provide beneficial effects for certain types of cancers are neither well-understood nor quite predictable.

Therefore, randomized clinical trials involving human subjects are imperative to ascertaining the efficacy of repurposed therapies for specific cancers, along with determining the optimal dosages necessary to achieve these significant therapeutic effects. Additionally, future research endeavors should prioritize elucidating the intricate interplay between CVD and cancer, thereby fostering the development of preventive strategies, and facilitating the co-management of cancer and CVD. This comprehensive approach is pivotal for ensuring successful cancer survivorship and optimizing patient outcomes in cardio-oncology.

## 5. Conclusions

In conclusion, the intricate nexus between CVD and cancer is increasingly acknowledged as a pivotal determinant of patient prognosis and therapeutic paradigms. Emerging evidence highlights a mutual, bidirectional relationship in which cancer and CVD distinctly influence one another’s outcomes. It is important to note that CVD does not increase the risk of cancer per se; rather, shared risk factors in patients with CVD may also promote cancer development. This convergence of disciplines presents formidable challenges alongside promising avenues for augmenting patient care. Recognizing the interconnected risk factors and molecular cascades between cancer and CVD, as well as the potential cardiotoxic ramifications of cancer therapies, is of paramount importance. The exploration of cardiovascular pharmacotherapeutics repurposed as adjunctive modalities in cancer management holds substantial potential, albeit remaining speculative and necessitating rigorous validation through randomized clinical trials. Moreover, adopting a holistic approach to the concurrent management of cancer and CVD is imperative for fostering resilient cancer survivorship and optimizing patient outcomes within the burgeoning domain of cardio-oncology. Hence, the advancement of our understanding pertaining to this complex interplay and the formulation of pre-emptive strategies stand as pivotal strides towards elevating the standard of healthcare delivery for individuals confronting the dual burden of cancer and cardiovascular disease.

## Figures and Tables

**Figure 1 cancers-16-01450-f001:**
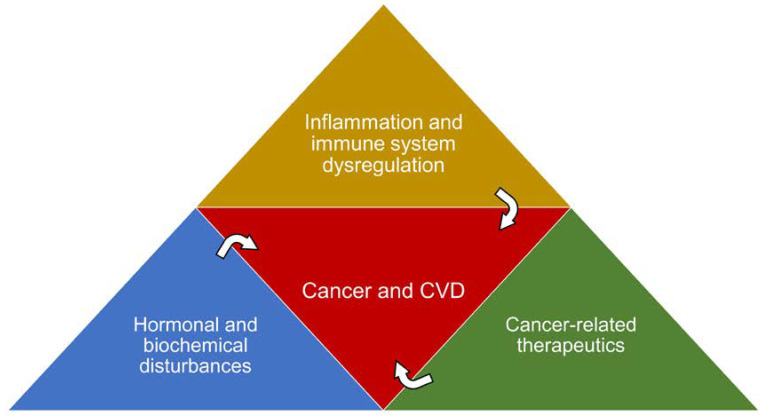
Illustration of shared pathophysiological pathways in cancer and cardiovascular disease (CVD) Cancer-related therapeutics include chemotherapy, radiotherapy, and immune checkpoint inhibitors.

**Table 1 cancers-16-01450-t001:** Potential desirable oncologic effects of commonly prescribed cardiovascular medications.

Medication	Potential Cancer Targets	Cancer Outcomes
Aspirin	CRC, esophageal, pancreatic, brain, lung, stomach, and prostate cancers	– Prevention of early neoplastic transformation and possible anti-metastatic effect – Mitigation of morbidity and mortality of some cancers
Angiotensin converting enzyme inhibitors (ACEi) Angiotensin receptor blockers (ARB)	Non-small-cell lung cancer Esophageal, pancreatic, and colon, prostate cancers and melanoma	– Reduction of incidence of some cancers and mitigation of metastatic burden – Enhance anticancer response prior to chemotherapy – Some observational data showed increased risk of lung cancer
Beta blockers (βB)	Breast, prostate, and ovarian cancers	– Prevention of certain types of cancers – Tumor growth reduction
Calcium channel blockers (CCB)	Breast, brain, colorectal, gastric, ovarian, and prostate tumors as well as leukemia	– Possible antiproliferative effects in preclinical studies
Statin	CRC, gastric, prostate and breast cancers	– Prevention of certain cancers. – Improvement of biochemical recurrence in prostate cancer
Proprotein convertase subtilisin kexin type 9 inhibitors (PCSK9-i)	Non-small-cell lung cancer, breast and colon cancers	– Improvement of anti-tumor immune response to cancer cells – Prevention of certain type of cancers – Tumor growth reduction and improved survival in certain cancers

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
