# Peer review of "Cancer and Cardiovascular Disease: The Conjoined Twins"

_cancers, 2024, doi:10.3390/cancers16081450_

Round 1
Reviewer 1 Report
Comments and Suggestions for Authors
Here, the authors propose a review of the interrelated aspects of cardiovascular disease (CVD) and cancer, two major causes of death worldwide. The authors discuss the shared risk factors, pathophysiological mechanisms, and therapeutic implications of CVD and cancer, and highlight the role of the cardio-oncology field in improving patient care and outcomes.
Major points:
The article covers a wide range of topics, from epidemiology and prevention to molecular biology and pharmacology, and provides relevant references and figures to support the arguments.
The article addresses a growing area of interest and importance, as the prevalence and survival of both CVD and cancer increase, and the need for multidisciplinary collaboration and personalized medicine becomes more evident.
The article proposes some interesting hypotheses and perspectives, such as the potential benefits of repurposing cardiovascular drugs for cancer treatment, and the possible role of metabolic reprogramming in both CVD and cancer pathogenesis.
Minor points:
The article mostly summarizes the existing literature, without providing a clear analysis or evaluation of the strengths and limitations of the evidence. The article also does not integrate the different topics or draw clear conclusions or recommendations. This should be discussed.
It would also be interesting to add a proposal on which direction, future research should take to improve cardiooncology research.
Author Response
We sincerely thank the reviewer for providing concise and constructive feedback. Their insights have proven valuable in refining the concluding section of our paper, particularly emphasizing the importance of prioritizing future research efforts toward elucidating the intricate interplay between cardiovascular disease and cancer.
Reviewer 2 Report
Comments and Suggestions for Authors
In this review, Dr. Zmaili and colleagues discussed the similarities and connection between CVD and cancer. I do have some comments to raise:
1) While CVD can be induced by cancers and cancer-related therapy, it is unclear how cancer can induced by CVDs. In lines 51-66, the authors only explain the "coincidence" based on epidemiology but it lacks pathophysiological explanations. Please add.
2) Figure 1 is not representative and it is unclear. Please use it to show the bidirectional connection between two diseases. Please revise
3) I am not sure if the pathogenesis of cancer in CVD is clear? Through which molecular mechanisms? This has to be clarified to say that they are interconnected.
4) Please illustrate the interlinked pathophysiology between those diseases.
5) Protooncogen and oncogen are important determinants in cancer development. How could CVD affect those?
6) In Table 1, please add the references to studies reporting that those drugs induce cancer. Otherwise this is only unproven speculation. Please also explain the mechanisms on how those drugs induce cancer. Logically, if they induce cancers, they would have been retracted and banned by the FDA. So I doubt this argument.
Author Response
Reviewer 2
Comments and Suggestions for Authors
In this review, Dr. Zmaili and colleagues discussed the similarities and connection between CVD and cancer. I do have some comments to raise:
1) While CVD can be induced by cancers and cancer-related therapy, it is unclear how cancer can induced by CVDs. In lines 51-66, the authors only explain the "coincidence" based on epidemiology but it lacks pathophysiological explanations. Please add.
We have revised our wording and included a conclusion section to provide clarification on these points.
2) Figure 1 is not representative and it is unclear. Please use it to show the bidirectional connection between two diseases. Please revise
Thank you for the suggestion. We added three main factors affecting both CVD and cancers simultaneously. We then drew the arrows to indicate the connection between these factors and both CVD and cancer.
3) I am not sure if the pathogenesis of cancer in CVD is clear? Through which molecular mechanisms? This has to be clarified to say that they are interconnected.
4) Please illustrate the interlinked pathophysiology between those diseases.
5) Protooncogen and oncogen are important determinants in cancer development. How could CVD affect those?
6) In Table 1, please add the references to studies reporting that those drugs induce cancer. Otherwise this is only unproven speculation. Please also explain the mechanisms on how those drugs induce cancer. Logically, if they induce cancers, they would have been retracted and banned by the FDA. So I doubt this argument.
Thanks for reviewing the table and for pointing this potential source of confusion. The table illustrates potential anticancer (not cancer-inducing) effects of these cardiovascular medications. To avoid the confusion, we added the word “desirable effects” to the title.
The table title “oncologic effects” not “oncogenic effects”
Reviewer 3 Report
Comments and Suggestions for Authors
The evolving landscape of cardio-oncology necessitates a nuanced approach to clinical trials, one that delves into the stratification of therapies based on an array of tumor characteristics, including cancer subtypes, genomic variants, and histological markers. This approach aims to transcend the limitations of studies that treat the "cancer" population as a monolithic entity, thereby uncovering differential responses to treatment that are intricately linked to specific disease features. Such a detailed stratification holds the promise of tailoring therapies more precisely, aligning treatment protocols with the unique aspects of each patient's cancer profile.
In parallel, there's a compelling need to design studies that explore the interaction between immune checkpoint inhibitors and cardio-oncology drugs. The hypothesis here is that combining these therapies may yield synergistic effects, potentially amplifying anti-tumor immune activity. The incorporation of mechanistic immunological assays into these studies will be pivotal in decoding the complex interplay between the immune system and cardio-oncology therapies, offering insights that could revolutionize treatment paradigms.
Furthermore, the optimization of treatment regimens through pharmacokinetic and dynamic research is paramount. Such research endeavors to establish the most efficacious dosing strategies, schedules, and formulation approaches, with a dual focus: maximizing the anti-cancer efficacy of cardiovascular medications and simultaneously curtailing off-target effects. This delicate balancing act underscores the necessity for meticulous research to harness the full therapeutic potential of these medications while safeguarding patient well-being.
In a holistic approach to treatment efficacy, the role of baseline lifestyle factors—diet, activity levels, sleep patterns, stress, and genomic polymorphisms—cannot be overstated. Integrating the analysis of these variables into clinical studies is crucial, aiming to quantify their influence on therapeutic response. This approach advocates for a more standardized or controlled lifestyle parameter within study protocols, thereby enabling a clearer understanding of how these factors intertwine with treatment outcomes.
Moreover, the economic dimensions of repurposing therapies in cardio-oncology warrant rigorous analysis. Conducting economic modeling studies to juxtapose the costs of these therapies against the quality-adjusted life years saved and cancer-attributable costs avoided could offer valuable insights into the fiscal prudence of these treatment avenues. Such analyses are integral in painting a comprehensive picture of the value proposition of repurposed therapies, guiding policy and clinical decision-making.
In the realm of precision medicine, leveraging bioinformatics to identify multiparametric biomarker signatures represents a frontier with immense potential. These biomarker signatures could serve as harbingers, predicting which patients stand to benefit most from cardio-oncology interventions. This predictive capability is a cornerstone in the shift towards more personalized, data-driven treatment approaches, ensuring that interventions are not just clinically effective but also aligned with individual patient profiles.
Furthermore, the integration of structured physical exercise programs as adjuvant treatments in cardio-oncology drug trials presents an intriguing prospect. This initiative seeks to evaluate the potential synergies between physical exercise and drug therapies, a convergence that might amplify therapeutic efficacy and patient quality of life. The structured nature of these exercise programs ensures consistency and measurable impact, providing a robust framework for assessing the interplay between physical activity and drug efficacy.
Lastly, the translation of findings from pre-clinical cell line and animal model studies to human trials is a critical phase in the research continuum. Initiating carefully planned and monitored phase 0/I first-in-human experiments prior to larger trials is instrumental. This step is a bridge between the promise of pre-clinical research and the reality of clinical application, ensuring that the transition is grounded in rigor, safety, and a profound understanding of the potential implications for human health. This phased approach not only underscores the commitment to evidence-based medicine but also embodies the ethos of patient-centric care, where every therapeutic stride is measured, scrutinized, and aligned with the overarching goal of enhancing patient outcomes in the realm of cardio-oncology.
Author Response
We express our sincere appreciation to the reviewer for providing invaluable insights into the field of cardio-oncology and offering valuable perspectives on future research directions. This impressive and thorough examination of the field's intricacies and recent scientific advancements greatly enriched our understanding and contributed significantly to the refinement of our work.
Reviewer 4 Report
Comments and Suggestions for Authors
The article is well written. The theme is interesting about the connection betweenn cancer and cardiovascular disease .the chapters are well developed. The figures are very illustarative for the purpose of the study. The refferneces are up to date. I recommend publication.
Comments on the Quality of English LanguageEnglish is ok.
Author Response
We express our sincere gratitude to the reviewer for their generous review and favorable recommendation.
Round 2
Reviewer 2 Report
Comments and Suggestions for Authors
Thanks for the response.
Comments on the Quality of English LanguageNo comment
Author Response
1. We have extensively discussed in the paper and there is also a schematic representation of the interlinked pathophysiology. Therefore, this is present in the paper.
2. This is a large topic and a manuscript in itself. Protooncogene and oncogene were not the aims of this paper and therefore discussion of this topic will deviate from the subject. But we do take under advisement that they do have a role to play.